# Physiological and Metabolic Changes in 'Xinyu Mandarin' Following Natural Tetraploidization

Yuting Wang, Shuilin Wan, Yuqing Tang, Huidong Yang, Chao Xu, Xincheng Liu , Zhongdong Hu * and Xinlong Hu *

Key Laboratory of Horticultural Plant Genetics and Physiology, Institute of Horticulture,
Jiangxi Academy of Agricultural Sciences, Nanchang 330200, China
* Correspondence: huzhongdong@jxaas.cn (Z.H.); huxinlong@jxaas.cn (X.H.)

**Abstract:** The mandarin is an important fruit crop worldwide, and 'Xinyu mandarin'—a local *Citrus reticulata* variety—is widely cultivated in Jiangxi Province, in China. Autopolyploidy has frequently been used for the improvement of crop varieties. In a previous study, we identified a tetraploid (4X) material of 'Xinyu mandarin' generated from its diploid (2X) mother seedling via a spontaneous mutation. However, the physiological and metabolic variations after polyploidy are not clear enough, which is not conducive to the early screening. In the present study, we analyzed the morphological, physiological, and metabolic differences between the tetraploid and diploid plants. The tetraploids had larger and thicker leaves, and the activities of key enzymes in photosynthesis, the total chlorophyll, phenolic, and proline contents, were all higher in tetraploids. In the non-targeted metabolomic profile, a total of 522 metabolites were identified, of which 61 were significantly different between diploids and tetraploids. The differential metabolites included similar proportions of primary and secondary metabolites, and most of these were up-regulated in tetraploids, especially stress-related metabolites such as phenolic acids, flavonoids, alkaloids, and so on, indicating that tetraploids may possess better stress tolerance ability and vigor. Therefore, the tetraploids of 'Xinyu mandarin' may serve as an excellent female parent for the improvement of citrus varieties.

**Keywords:** polyploidy; *Citrus reticulata*; plant metabolites; autotetraploidy; adaptation; disease resistance

## 1. Introduction

Polyploidy is ubiquitous in plants, has profound implications for their evolution and ecology, and is considered a common method of speciation [1,2]. Polyploidy is an irreversible process that results from genome-wide replication, and it occurs more frequently in plant species [3]. The polyploidy-induced modulation of gene expression and epigenetic remodeling of polyploid genomes cause physiological and morphological changes in plants [4]. There are two types of polyploids: autopolyploids and allopolyploids [5]. With the dual advantages of polyploidy and hybridization, allopolyploids can better cope with evolutionary changes and show increased species survival, which is important for the long-term evolution and improvement of crops [6,7]. Moreover, growing evidence suggests that autopolyploids may be more common in nature than expected. Therefore, polyploidy in plants has gradually attracted widespread research attention.

Autopolyploidization refers to the addition of a chromosomal set from the same species, and it mostly occurs during whole-genome duplication [8]. Compared with allopolyploids, autopolyploids may be more adaptable to environmental changes in a short period of time through alterations in their metabolic phenotype [9]. Autopolyploidization can lower the uncertainty associated with allopolyploidy, is more helpful for strengthening plant reproduction, and improves plant colonization and distant hybridization [10]. Autopolyploidization has been widely used in plant breeding—including in species such as citrus, potatoes, watermelons, and blueberries—to obtain favorable traits and shorten breeding cycles [11–14].

Autopolyploidization has complex effects on the mutagenicity of plant organs, which is closely related to the cytological structure and dynamic growth pattern of each organ [15]. Autopolyploidization leads to the generation of cells with increased plasticity and stability [16,17]. In addition, it can also cause changes in the size of the nucleus, cell, organ, and organism; as such, autopolyploidization usually results in the enlargement of single organs or even the entire plant [18,19]. Autopolyploidization affects both primary and secondary metabolism to varying degrees and is a promising approach for enhancing the biosynthesis of plant secondary metabolites [20,21]. At the same time, it also improves drought resistance and salt tolerance in plants [22,23]. Hence, autopolyploidization is an important tool for contemporary commercial breeding [24].

There are several important ways autopolyploidy can be induced. It can be naturally occurring, but it can also occur by artificial induction, somatic cell fusion, and so on [14]. Although naturally occurring autopolyploid plants generally tend to be taller and show stronger stress tolerance, autopolyploidy is extremely rare in natural populations [25]. The incidence of tetraploidy in polyembryonic citrus varieties ranges from 1% to 5.6% [26], which has limited the applications for breeding. Therefore, artificial methods (such as treatments with chemical and physical agents) are always used to increase the frequency of polyploidy in plants [27]. Colchicine treatment was proved to produce a high frequency of autopolyploid induction, and it was applied to citrus [28,29]. However, artificially induced polyploidy often results in variable plant morphology and poses potential risks related to decreased fertility, slow growth, and dwarfism. As a result, artificially induced polyploidy is associated with a high pre-screening burden before an optimal strain can be obtained for breeding [30]. In addition, somatic cell fusion is widely used in citrus breeding because of its ability to hybridize species and create unique germplasm. Over the past few decades, breeders have used cell fusion techniques to create more than 200 citrus polyploids and developed some excellent varieties [31–33]. However, this method has relatively high technical requirements and expensive equipment and can only be carried out in the laboratory. Subsequently, in order to make efficient use of these created polyploids, developing rapid and accurate pre-screening methods could play a vital role. However, the metabolic changes induced by natural autopolyploidization—and the mechanisms underlying these changes—remain to be elucidated.

Citrus is an important fruit crop that is cultivated worldwide [34], and plants in this genus are native to China. Southern China is particularly rich in citrus resources and has a long history of cultivation of indigenous citrus species [35]. The 'Xinyu mandarin' (*C. reticulata*) is a local citrus fruit in Jiangxi Province, and it has bright color and good texture and taste. However, its development has been limited by seediness, and this phenomenon also exists in many other citrus species. However, the commercial development of citrus had aimed at seedless breeding. At present, hybrid breeding by autotetraploidy is one of the main breeding tools for obtaining kernel-free citrus fruits. We previously found a tetraploid strain of *C. reticulata* that exhibits spontaneous mutation and can enter the fruiting period. To further analyze the differences between naturally tetraploids and diploids and to test whether this tetraploid had the characteristics to be a good parent, we compared the biological and metabolic characteristics between tetraploids and diploids to evaluate the key metabolic differences, which may provide more indicators for the screening of polyploid citrus varieties.

## 2. Materials and Methods

### 2.1. Plant Materials

Diploid *C. reticulata* (C2X) were authenticated and collected from the Germplasm Resource Nursery (Nanchang, China; annual rainfall 1600–1700 mm, RH 78.5%, min −15.7 °C, max 40.9 °C, and average temperature 17.7 °C) of the Institute of Horticulture at Jiangxi Academy of Agricultural Sciences. Seedlings of tetraploid plants (C4X) were derived from the diploids (C2X) in 2015 and planted in the above nursery. Diploids (C4X) had similar growth period and culture conditions to C2X and were authenticated and collected from

the above nursery. The plant variety was identified by two senior researchers, Wan Shuilin and Hu Zhongdong, at the aforementioned institute. The scions from the C4X were collected and grafted in the high position to the C2X plant in 2015. Both of them appeared to grow well.

### 2.2. Sampling and Morphological Analysis

All the leaves in this experiment were collected from the grafted branches and their neighbors in the same orientation on the plant. The fourth or fifth leaves counted from the apex of the current-year spring flush were collected before 9 a.m. on 31 May 2019. The new leaves that had just come out (2–3 leaves for each sample) were used for ploidy analysis using flow cytometry (Partec CyFlowspace, Shanghai, China). The fresh leaves (10 leaves for each sample) were measured by electronic micrometer to calculate leaf index (length/width), while the thickness was measured in the meanwhile. After that, the leaves were initially ground and mixed with liquid nitrogen in a mortar and stored at $-80\,^\circ$C and used for physiological and metabolite profiling. Each experiment was conducted in triplicate from three different branches.

### 2.3. Determination of Physiological Indices

The fresh leaves (0.2 g per sample) were extracted with 80% acetone and then dissolved in 95% alcohol; the chlorophyll a and chlorophyll b content was estimated by measuring the absorbance at 665 and 649 nm [36].

The activities of key photosynthetic enzymes, including ribulose diphosphate carboxylase/oxygenase (Rubisco) and pyruvate phosphate dikinase (PPDK), phenolic compounds, lignin, and proline in leaves (0.2 g per sample) were measured using assay kits (Suzhou Keming Biotechnology Co., Ltd., Suzhou, China).

Fresh leaves, previously crushed, were freeze-dried (48 h). The 100 mg freeze-dried sample was accurately weighed, and 1.0 mL of each different extract solution was added to determine lignin, total phenol, and proline. Rubisco activity was estimated by measuring the rate of oxidation of reduced coenzyme I (NADH) at 340 nm, and PPDK activity was calculated by measuring the rate of reduction of reduced coenzyme I at 340 nm [37].

The reduction of the tungsten–molybdenum complex by phenolic compounds produces a blue compound with a characteristic absorbance peak at 760 nm. The total phenolic content was calculated by measuring absorbance at this wavelength [38]. Following acetylation, the phenolic hydroxyl group in lignin has a characteristic absorbance peak at 280 nm, and the lignin content was calculated by measuring the absorbance at this wavelength. The sample was extracted with sulfosalicylic acid and then reacted with an acidic ninhydrin solution. After toluene extraction, absorbance at 520 nm was measured to calculate the proline content.

To determine the endogenous hormone levels, mixed leaf samples were ground into powder with a freezer grinder (MM400, Retsch, Haan, Germany). Following this, 1 mL of the extraction solution (methanol: water: formic acid; 15:4:1; *v/v/v*) was added to 52 mg of the ground sample, and the mixture was ultrasonically extracted for 20 min in an ice bath. The supernatant was obtained by centrifugation ($6367\times g$, 10 min, 3 times), filtered (20 μm), dried with nitrogen, and dissolved in 200 μL methanol for LC–MS/MS analysis. An HSS T3 (C18) chromatographic column (ACQUITY UPLC, Waters Corp., Massachusetts, USA; dimensions, 2.1 mm × 100 mm; particle size, 1.8 μm) temperature was maintained at 35 $^\circ$C (flow rate, 0.4 mL/min). The mobile phase was set as follows: distilled water/acetonitrile (90:10 *v/v*) at 0 min, 85:15 *v/v* for 10 min, 60:40 *v/v* for 20 min, and 20:80 *v/v* at 22 min.

The MS analysis (LCMS-8050, Shimadzu, Kyoto, Japan) was performed with the following conditions: source/desolvation temperatures, 300 $^\circ$C/400 $^\circ$C; cone/desolvation gas, 180/1200 L/h; spray voltage, 3.5 kV; cone voltage, 10–40 V; collision energy, 12–30 eV [39].

### 2.4. Non-Targeted Metabolic Profiling

All leaf samples were lyophilized with a freezer dryer (VIRTIS Freeze Mobile 25 L, Shanghai, China) for 48 h and ground with an automatic grinder and zirconia beads for 1.5 min at 30 Hz. The ground sample (100 mg) was mixed with 0.6 mL of 70% aqueous methanol and left overnight at 4 °C. Finally, the powder was obtained by centrifugation at $10,611 \times g$ for 10 min. A 0.5 mL aliquot of the supernatant was passed through a solid-phase extraction column (CNWBOND Carbon–GCB SPE Cartridge, ANPEL, Shanghai, China; 250 mg/3 mL). The sample was filtered through a filter membrane (SCAA–104, ANPEL, Shanghai, China; pore size, 0.22 μm) and used for Ultra Performance Liquid Chromatography–Electrospray Ionization–Tandem Mass Spectrometry (UPLC–MS/MS) analysis with a UPLC-ESI-MS/MS system (UPLC: Shim-pack UFLC Shimadzu CBM30A, Kyoto, Japan; MS: Applied Biosystems 6500 Q TRAP, Foster City, CA, USA).

UPLC analysis was performed using an HSS T3 (C18) chromatographic column (Waters Corp., Massachusetts, USA; dimensions, 2.1 mm × 100 mm; particle size, 1.8 μm). The mobile phases consisted of 0.04% acetic acid in water (A) and 0.04% acetic acid in acetonitrile (B) and were used for gradient elution as follows: 0.0–10.0 min (5% B–95% B), 10.0–11.0 min (95% B–95% B), 11.0–11.10 min (95% B–5% B), and 11.0–14. 0 min (5% B–5% B) at a flow rate of 0.3 mL/min. A 4 μL aliquot of the sample was injected. The column temperature was set to 40 °C, and this system was connected to the ESI–Triple quadrupole linear ion trap (QTRAP)-MS system.

The conditions for mass spectrometry were as follows: electrospray ion source (ESI) temperature, 550 °C; mass spectrum voltage, 5500 V; curtain gas, 30 psi; collision-activated dissociation parameters set to 'high'. In the triple quadrupole tandem (QQQ) system, each ion pair was scanned and detected according to the optimized decluttering potential and collision energy.

### 2.5. Statistical Analysis

(1) Principal component analysis (PCA), hierarchical cluster analysis (HCA), and calculation of Pearson's correlation coefficients (PCC): The data were scaled to unit variance and included in an unsupervised PCA utilizing the "prcomp" package in R statistical software (www.r--project.org, URL (accessed on 11 November 2019)). HCA and the PCC analysis were performed in R, and the results were presented as heatmaps.

(2) Selection of differential metabolites: Metabolites that were significantly down-regulated or up-regulated between groups were determined based on a variable importance in projection (VIP) value of $\geq 1$ and an absolute log2 fold change (FC) value of $\geq 1$. VIP values were extracted from an orthogonal partial least squares discriminant analysis (OPLS–DA), and the score plots and permutation plots were obtained using the "MetaboAnalyst" package in R. The data were log-transformed (log2) and mean-centered before OPLS-DA. A total of 200 permutations were performed to avoid overfitting.

(3) KEGG annotation and pathway enrichment analysis: The KEGG Compound database was used to annotate the metabolites (http://www.kegg.jp/kegg/compound/, URL (accessed on 12 December 2019)), and the annotated metabolites were mapped to the KEGG Pathway database (http://www.kegg.jp/kegg/pathway.html, URL (accessed on 12 December 2019)). Pathways with significantly up- or down-regulated metabolites were fed into a metabolite sets enrichment analysis. Significance levels were determined by a hypergeometric test.

(4) The SAS software (version 9.1, SAS Institute Inc., NC, USA) was used for statistical analysis. Differences in hormone levels were analyzed with a Student's *t*-test, and significance was determined at 0.01.

## 3. Results and Discussion

### 3.1. Changes of Morphology following Natural Tetraploidization

Spontaneous mutation can result in a wide range of phenotypic changes. To verify that the spontaneous mutants were homozygous polyploids, we performed ploidy analysis

in two samples using flow cytometry. Compared with the diploid mother (2X; Figure 1a), the spontaneous mutant was confirmed to be a tetraploid (Figure 1b). The morphological similarities between the samples confirmed that the spontaneous mutant was an autotetraploid (4X). Genome doubling is generally believed to cause morphological changes. It was shown that the autotetraploids had larger and thicker leaves and lower leaf index (Figure 1c,d). This result is consistent with previous results [18,19]. It was reported that there was a significant positive correlation between plant genome size and epidermal cell area. Thus, polyploidy may increase cell size by increasing the nuclear DNA content [40].

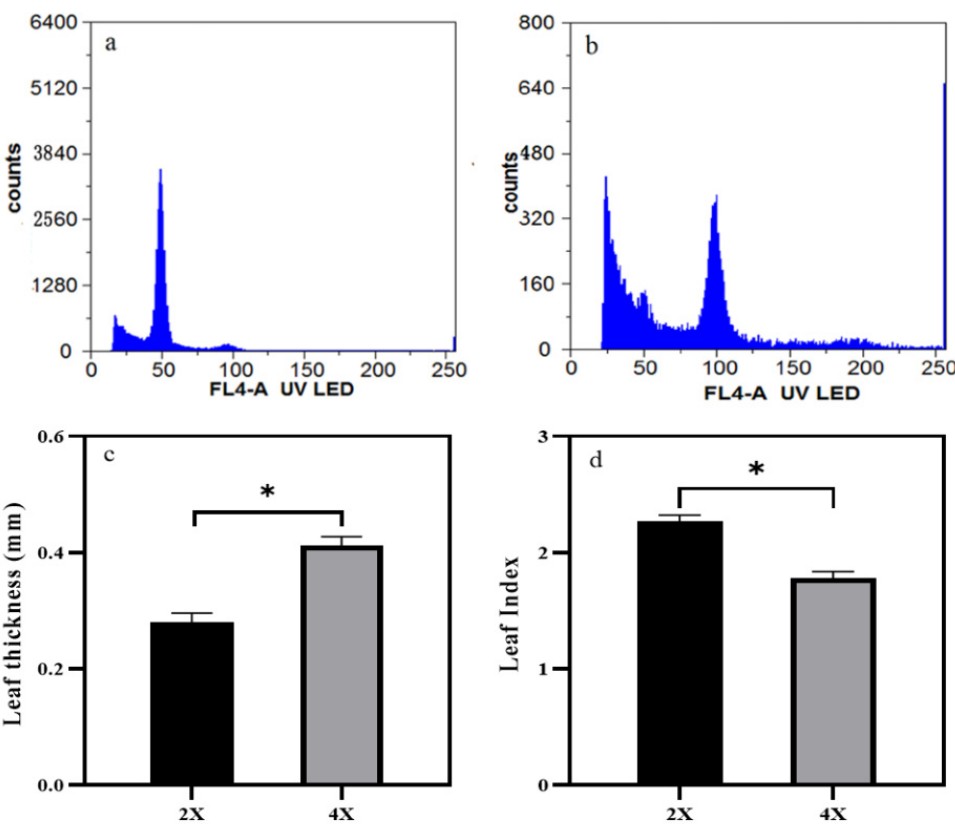

**Figure 1.** Ploidy analysis and comparison of the shape of leaf cells. (**a**) Normal diploid (2X) cell; (**b**) natural tetraploid (4X) cell; (**c**) leaf index; (**d**) leaf thickness in diploid and tetraploid leaves. * was the significance determined at $p < 0.01$ ($n = 10$).

*3.2. Changes of Physiological Indices following Natural Tetraploidization*

Genomic changes lead to important changes in the physiological and biochemical functions of plants. We analyzed the morphological differences between diploids and autotetraploids to evaluate the physiological and biochemical indices of the autotetraploids. Plant growth is accompanied by continuous protein synthesis in the cells of the aerial parts, which provides the building blocks for new cells, tissues, and organs. The protein content of 4X plants was significantly higher than that of 2X plants (Figure 2a), indicating that the metabolism of tetraploid plants may be more vigorous than that of diploids. Polyploidy can affect the growth activity of plants, but it varied between the autopolyploidy and the allopolyploidy. The growth activity of homologous polyploids is usually weaker than that of diploids, possibly because of the reduced cell division ability [41]. However, the heteropolyploid showed the opposite phenomenon, possibly due to heterosis [42].

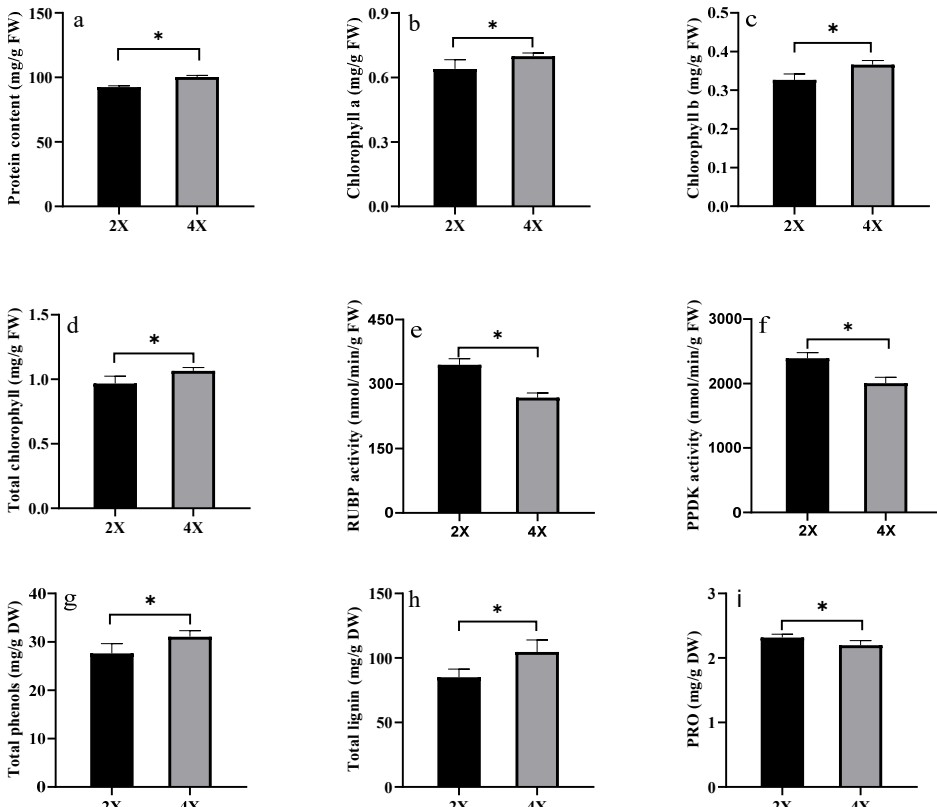

**Figure 2.** The physiological and biochemical indices of leaves obtained from diploid and tetraploid plants (**a**–**i**, protein content, chlorophyll a, chlorophyll b, total chlorophyll, activity of Rubisco, activity of pyruvate phosphate dikinase (PPDK), total phenolics level, lignin level, proline content). * was the significance determined at $p < 0.01$ ($n = 3$).

Photosynthesis is an important function of leaves, and chlorophyll a and b play key roles in this process. The chlorophyll a and b (Figure 2b,c) contents of leaves reflect their photosynthetic capacity. In addition, Rubisco and PPDK are key enzymes involved in photosynthesis. Samples obtained from the 4X plants had higher chlorophyll content (Figure 2b–d) and showed stronger Rubisco and PPDK activities (Figure 2e,f) than samples obtained from 2X plants (Figure 2), indicating that the autotetraploid plants had higher photosynthetic capacity. The similar phenomenon has been demonstrated in the autotetraploid 'Hanfu' apple compared with its diploid [42].

Stress tolerance is an important quality attribute in crops, as it provides tolerance to adverse environmental conditions (including drought, disease, cold, and salt stress). Numerous studies have shown that tetraploids demonstrate excellent anti-stress resources [1,22,23]. The total polyphenol and lignin levels of plants are related to their disease resistance. Total polyphenol and lignin levels were higher in tetraploids than in diploids, which implied that 4X plants may have enhanced resistance to disease, whereas the proline content was lower (Figure 2g–i). It was reported that the resistance of autotetraploidy was related to genetic background. The varieties that were disease-resistant had shown enhanced tolerance of diseases after autohybridization, whereas the sensitive varieties did not change significantly [43]. Since 'Xinyu mandarin' was proved to be resistant to disease in production, its autotetraploid should have even stronger resistance.

Hormones are important compounds that are induced by plan t cells to receive specific environmental signals and are involved in the regulation of plant growth and development. We had determined several kinds of main hormone components (GA20, GA24, IAA, ICAld JA, JA–ILE SA) in the leaves of tetraploids and diploids; there are four significantly different kinds shown in Table 1. Compared to the diploid, abscisic acid and

isopentenyl adenine were higher in the tetraploid. In addition to inhibiting growth, abscisic acid can also regulate stomatal closure and enhance stress tolerance, which may be the result of higher content of polyphenolic lignin, a stress-resistant component, in tetraploid leaves [22,23,44]. Genome doubling could affect hormone levels and the expression of stress-related genes [45]. Isoamyl alkenyl adenine promotes the formation of chlorophyll and accelerates plant metabolism and protein synthesis, which corresponds to the results of higher chlorophyll and protein content and stronger activity of key enzymes in photosynthesis in tetraploid leaves. Indoleacetic acid was not detected in diploid leaves, while tetraploid leaves reached about 4 ng/g, which may be due to the vigorous growth after genome doubling and the inhibition of excessive growth by high concentration of indoleacetic acid. Trans-zeatin content in the leaves of the tetraploid was significantly lower than that of the control diploid, indicating that the growth metabolism was enhanced after genome doubling, while the differentiation energy was reduced through IP. This indicates that the spontaneously mutated tetraploid is in potentially excellent shape, with high stress tolerance and vigorous growth.

**Table 1.** The content of 10 common hormones in two samples (ng/g, FW).

| Hormone Component | Group | |
|---|---|---|
| | 'Xinyu Mandarin' 2X | 'Xinyu Mandarin' 4X |
| Abscisic acid/ABA | $12.97 \pm 1.16$ | $20.5 \pm 2.00$ * |
| 3-Indoleacetic acid/IAA | N/A | $4.29 \pm 0.26$ |
| N6-(2-Isopentenyl) adenosine/IP | $0.0270 \pm 0.0007$ | $0.0381 \pm 0.0014$ * |
| BioReagent/tZ | $0.221 \pm 0.012$ | $0.154 \pm 0.003$ * |

Note: The data in the table are the mean $\pm$ standard error of three biological repeptitions. The significance of difference between 4X and 2X was analyzed by Student $t$ test. * was the significance determined at $p < 0.01$ ($n = 3$).

### 3.3. Non-Targeted Metabolic Analysis

To clarify the metabolic consequences of tetraploids after natural mutation, we performed a comparative non-targeted metabolomic analysis of tetraploids using normal diploids as control. Following HPLC–MS/MS analysis, a total of 522 metabolites were identified, including 174 flavonoids, 68 lipid compounds, 60 phenolic acid compounds, 59 amino acids and their derivatives, 34 organic acids, 32 nucleotides and their derivatives, and 28 alkaloids (Figure 3a). We assessed the metabolite data using HCA, and the results (Figure 3b) revealed clear differences between the metabolic profiles of the two samples (4X vs. 2X). An unsupervised PCA clearly separated the two experimental samples (2X and 4X) from the quality control (QC) samples (Figure 3c). In addition, the QC group had a high concentration of data points, indicating high reproducibility of the collection process.

The separation of the two sets of samples into different regions based on the first (PC1) and second (PC2) principal components highlighted the effect of genome doubling on the metabolite profiles of the samples. Taken together, PC1 (44.59%) and PC2 (11.25%) explained 55.84% of the total variance in the data. Samples from the tetraploid and diploid seedlings were considerably separated, indicating that genome doubling induced a genome–wide diversification of metabolites in *C. reticulata*.

To further eliminate anomalies in pattern recognition, the metabolite data were log10–transformed and subjected to HCA to clarify the similarity and degree of similarity in metabolites between samples. The results of HCA revealed clear differences between tetraploids and diploids (Figure 4a). The OPLS–DA model was used to analyze significant differences between samples and to screen for differential metabolites between tetraploids and diploids. The score plot of the OPLS–DA model (Figure 4b) indicated that the model was appropriate and revealed a significant difference in metabolites between tetraploid and diploid *C. reticulata*.

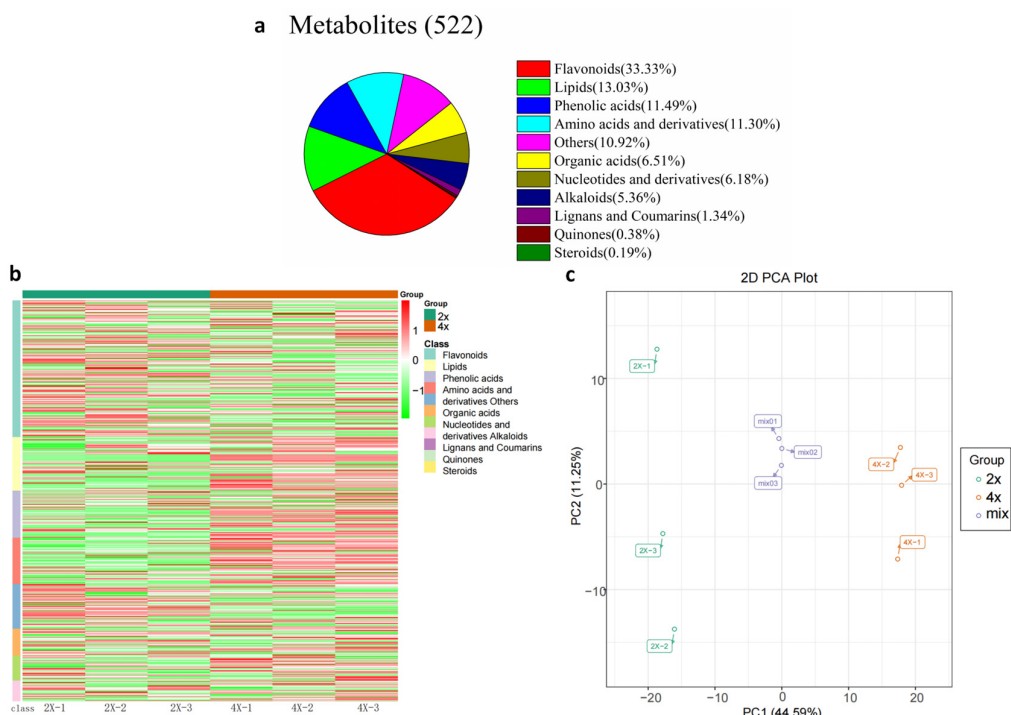

**Figure 3.** Assessment of the quality of metabolomic data obtained from diploid (2X) and tetraploid (4X) samples. (**a**) Types and proportions of the metabolites; (**b**) results visualized as a heatmap; (**c**) results of principal component analysis. Equal volumes of 2X and 4X samples were mixed and used as a quality control (QC).

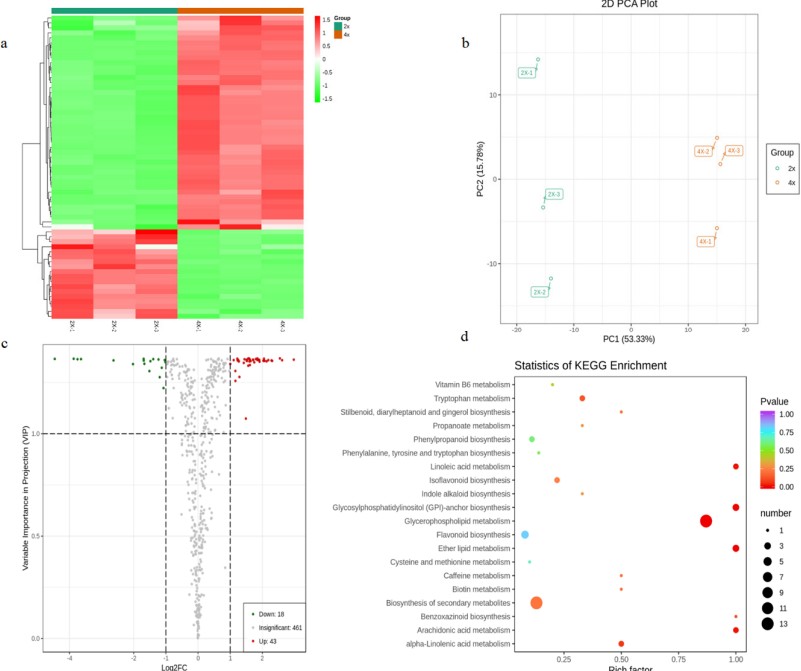

**Figure 4.** Multivariate analysis of identified metabolites. (**a**) Hierarchical clustering analysis of metabolites identified from diploid (2X) and tetraploid (4X) samples; (**b**) plot of the orthogonal partial least squares discriminant analysis model of metabolites identified from 2X and 4X samples; (**c**) volcano plot of 61 differential metabolites; (**d**) overview of the KEGG pathway analysis of differential metabolites.

To accurately identify differential metabolites, univariate analysis is often followed by multivariate analysis. Here, we screened the differential metabolites by analyzing the differences in FC (univariate analysis) and VIP values (from a multivariate analysis of the OPLS–DA model). Metabolites with FC $\geq$ 2 and VIP $\geq$ 1 were considered differential metabolites between tetraploids and diploids. A total of sixty-one differential metabolites were identified in sixteen different categories, including lipids (twenty-four), phenolic acids (eleven), flavonoids (eight), alkaloids (four), organic acids (four), amino acids and their derivatives (two), vitamins (two), sugars (two), and others (four). The KEGG pathway enrichment analysis of these differential metabolites (Figure 4d) revealed that they were mostly involved in glycerophospholipid metabolism, biosynthesis of secondary metabolism, ether lipid metabolism, and glycosylphosphatidylinositols anchor biosynthesis.

*3.4. Changes of Metabolism following Natural Tetraploidization*

Metabolites are the material manifestation of the life activities of plants in different developmental stages and are classified into two categories: primary and secondary metabolites [46,47]. To further discover the metabolic variation induced by autotetraploidization, we analyzed the primary and secondary metabolites of two materials, respectively, using differential analysis and KEGG enrichment analysis.

Primary metabolism is the process through which plant cells provide energy and intermediate products for plant survival, growth, development, and reproduction [48]. Common primary metabolites include sugars, amino acids, and nucleic acids. A total of 30 differential primary metabolites were identified (Table 2), including organic acids (four) and amino acids and their derivatives (two), which was similar to that of three citrus double diploids [19]. However, lipids (24) were most prevalent in the 4X pants, which may be due to variety specificity. Lipids, as one of the most important metabolites in plants, were up–regulated in tetraploids. This may be related to the increase in cell size, changes in cell membrane composition, and increase in lipid levels after genome doubling [49,50]. Only two types of amino acids and their derivatives were up-regulated in tetraploids. These changes may have negligible effects, as amino acids and their derivatives accounted for a very small proportion of all differential metabolites. Organic acids not only participate in the transfer of electrons and protons in metabolic reactions, but also act as transporters of redox capacity [51]. Although chlorophyll levels and the activities of key enzymes in photosynthesis were higher in tetraploids, the organic acid content of tetraploids decreased overall. This may be due to a restriction of metabolic reactions or compensation for the enhanced photosynthetic capacity of tetraploids, which may help maintain photosynthetic activity at a reasonable level. Of these different metabolites, the majority were up-regulated. This suggests that genome doubling may stimulate the primary metabolism [19,47,52].

Secondary metabolites are natural compounds that plants synthesize from primary metabolites during specific periods of development. These compounds promote plant adaptation to adverse environments, improve pathogen resistance, and regulate plant metabolism [52]. Secondary metabolites can be divided into three main categories: nitrogen-containing organic compounds, terpenoids, and phenolic compounds. In total, we identified thirty-one types of differential secondary metabolites (Table 3), including twelve types of phenolic acids, eight types of flavonoids, four types of alkaloids, and seven other types of metabolites.

**Table 2.** Representative differential primary metabolites.

| Compounds | Fold Change | Type | Compounds | Fold Change | Type |
|---|---|---|---|---|---|
| Amino acids and derivatives | | | | | |
| 4-Hydroxy–L–glutamic acid | 2.24 | up | L–Homocystine | 2.44 | up |
| organic acid compounds | | | | | |
| (S)–2–Hydroxybutanoicacid | 0.07 | down | 5–Hydroxyhexanoic acid | 0.31 | down |
| α–Hydroxyisobutyric acid | 0.08 | down | 2–Hydroxybutanoic acid | 0.07 | down |
| lipid compounds | | | | | |
| 1–Stearoyl–sn-glycero–3–phosphocholine | 4.55 | up | LysoPC 17:0 | 4.10 | up |
| LysoPE 18:1 | 4.20 | up | LysoPC 14:0(2n isomer) | 3.06 | up |
| LysoPC 18:3 | 2.35 | up | LysoPC 16:0(2n isomer) | 3.36 | up |
| LysoPC 16:0 | 3.29 | up | LysoPC 18:0 | 4.95 | up |
| LysoPE 18:1(2n isomer) | 4.10 | up | LysoPE 16:0(2n isomer) | 3.77 | up |
| LysoPC 16:2(2n isomer) | 2.87 | up | LysoPC (18:2) | 3.59 | up |
| LysoPE 14:0 | 3.06 | up | LysoPC (18:2) isomer | 3.75 | up |
| LysoPC 18:3(2n isomer) | 2.39 | up | LysoPC(16:1) | 2.86 | up |
| LysoPE 18:2(2n isomer) | 6.12 | up | LysoPC(18:2) | 3.68 | up |
| LysoPE 16:0 | 3.69 | up | LysoPC(16:0) | 3.11 | up |
| LysoPC 15:1 | 2.68 | up | LysoPC(18:1) | 3.03 | up |
| LysoPC 15:0 | 2.85 | up | LysoPC(18:0) | 4.87 | up |

Note: All metabolites are listed with the fold change is greater than twice or less than half in tetraploid *C. reticulata* compared to the diploid.

**Table 3.** Representative differential secondary metabolites.

| Compounds | Fold Change | Type | Compounds | Fold Change | Type |
|---|---|---|---|---|---|
| Phenolic acid compounds | | | | | |
| Di–p–Coumaroyltartaric acid | 0.44 | down | p–Coumaroylferuloyltartaric acid | 0.31 | down |
| Neochlorogenic acid (5–O–Caffeoylquinic acid) | 5.84 | up | 3–O–(E)–p–Coumaroyl quinic acid | 2.33 | up |
| Chlorogenic acid | 2.72 | up | 6′–trans–Cinnamoyl–8–epikingisidic acid | 4.14 | up |
| Chlorogenic acid methyl ester | 2.41 | up | Trans-3–O–p–coumaric quinic acid | 3.01 | up |
| Cryptochlorogenic acid | 7.88 | up | Cis–3–p–coumaric quinic acid | 3.59 | up |
| 3–O–Feruloyl quinic acid | 2.94 | up | Scopoletin(7-Hydroxy-5-methoxycoumarin) | 2.80 | up |
| Flavonoids | | | | | |
| Naringenin–7–O–glucoside | 2.24 | up | 5,6,7,8,3′,4′–Hexamethoxyflavanone | 0.43 | down |
| 6–C–Hexosy–apigenin O–feruloylhexoside | 0.25 | down | Apigenin–7–O–(6–O–Malonyl Glucoside) | 0.36 | down |
| Kaempferol 7–O–rhamnoside | 0.31 | down | Apigenin–7,4′–dimethylether | 0.38 | down |
| Apigenin 7–rutinoside (Isorhoifolin) | 0.49 | down | Linarin | 0.16 | down |
| Alkaloids | | | | | |
| Tryptamine | 2.95 | up | Indole | 2.16 | up |
| 3–{(2–Aminoethoxy) (hydroxy) phosphoryl] oxy}–2–12–octadecadienoate | 4.39 | up | 3–{(2–Aminoethoxy) (hydroxy) phosphoryl] oxy}–2–hydroxypropyl palmitate | 3.47 | up |
| | | | Others | | |
| Xanthosine | 2.00 | up | Propyl2–(trimethylammonio) ethyl phosphate | 3.27 | up |
| 1,1–Kestotetraose | 0.48 | down | Biotin | 0.35 | down |
| D–(+)–Melezitose | 0.46 | down | Limonin | 0.05 | down |
| Pyridoxine | 0.49 | down | | | |

Note: All metabolites are listed with the fold change is greater than twice or less than half in tetraploid *C. reticulata* compared to the diploid.

Phenolic compounds are generally considered to be important in plant resistance to environmental stress [53]. Almost all up-regulated phenolic compounds may be indicative

of increased stress tolerance in tetraploid *C. reticulata*. However, as one of the important phenolic compounds, flavonoids are mostly affected by light [54]. Therefore, the down-regulation of these phenolic compounds may indicate that light factors affect polyploids to a lower extent than other environmental factors [52]. Alkaloids are produced as a result of pathogen stress [55], and the four types of alkaloids identified as secondary differential metabolites were all up-regulated in tetraploids. This suggests that genome doubling may enhance disease resistance in tetraploid *C. reticulata*.

Compared with the diploid control, the naturally mutated tetraploids had larger and thicker leaves. The analysis of physiological and biochemical indices revealed that the tetraploids showed better photosynthetic capacity and up-regulated hormones than 2X; moreover, the secondary metabolites related to stress, such as phenolic acids, flavonoids, and alkaloids, were also proved to be up-regulated. In this case, the tetraploids of 'Xinyu mandarin' could be exhibiting high stress tolerance and vigorous growth.

In the non-targeted metabolomic profile, key metabolic differences between tetraploid and diploid 'Xinyu mandarin' were analyzed. A total of 522 metabolites were identified. Of these, 61 (11.7%) were significantly different between diploids and tetraploids, indicating that overall, natural genome doubling did not have a drastic effect on metabolism in *C. reticulata*. It indicated that there is little variation in gene expression between homologous tetraploid and diploid parents, which is consistent with many studies [19,45]. Moreover, the differential metabolites included primary and secondary metabolites in similar proportions, and most of these were up-regulated. In contrast, the organic acid content was lower in tetraploids, likely as a compensation for enhanced metabolism. It accords with the model proposed by predecessors [19]. Among secondary metabolites, compounds associated with plant adaptation to short-term environmental changes—such as phenolic acids and alkaloids—were up-regulated in tetraploids, whereas light-related flavonoids were down-regulated. This indicated that tetraploidy may help maintain the original biological characteristics of a diploid, while simultaneously allowing the plant to develop a stronger potential for adaptation to the current environment. However, the expression of related varied metabolites must still be verified.

## 4. Conclusions

Autopolyploidy has always been used for the improvement of crop varieties. Here, we performed a comparative analysis of a spontaneously mutated 'Xinyu mandarin' tetraploid (4X) and its diploid mother (2X). The tetraploid of the 'Xinyu mandarin' maintained basic characteristics of the diploid and showed changes such as increased resistance in morphology and physiology. Most of the significantly different metabolites were up-regulated in tetraploids, especially stress-related metabolites such as phenolic acids, flavonoids, alkaloids, and so on, indicating that the tetraploids may possess better stress tolerance ability and vigor. Therefore, tetraploid *C. reticulata* may serve as an excellent female parent for the genetic improvement of *C. varieties* with potential commercial value. Meanwhile, these differential metabolites would provide an alternative reference for the evaluation of citrus varieties, and the results have enriched the study involved with polyploidization.

**Author Contributions:** The study was conceived and designed by Y.W. and S.W.; Data acquisition was performed by Y.W., C.X., X.L. and H.Y.; Data analysis and interpretation were performed by Y.W. and Y.T.; The draft manuscript was revised critically for intellectual content by Z.H. and X.H. All authors have read and agreed to the published version of the manuscript.

**Funding:** This work was supported by the Modern Agricultural Research Collaborative Innovation Program of Jiangxi Province (JXXTCXQN202106, JXXTCX202203), the Earmarked Fund for the China Agriculture Research System (CARS-26), the Earmarked Fund for the Jiangxi Agriculture Research System (JXARS-07), and the Key Research and Development Program of Jiangxi Province (20202BBFL63015).

**Institutional Review Board Statement:** The studies was not involving humans or animals.

**Data Availability Statement:** The authors declare that the data associated with this paper are available and will be publicly accessible upon publication. The DOI for the dataset is 10.6084/m9.figshare.21435297.

**Conflicts of Interest:** The authors declare no conflict of interest.

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
