# Peer review of "Physiological and Metabolic Changes in ‘Xinyu Mandarin’ Following Natural Tetraploidization"

_agronomy, doi:10.3390/agronomy13010029_

Round 1

Reviewer 1 Report

I have thoroughly reviewed the submitted paper ‘Physiological and Metabolic Changes in ‘Xinyu mandarin’ Following Natural Tetraploidization’ which deals with an interesting topic.

In this, authors have compared the performance of a natural Tetraploid with respect to the diploid Citrus reticulata on the basis of various important leaf morphological, physiological, and metabolic processes. The study was planned and executed well and also presented nicely, reporting interesting findings with describing how tetraploid is better than diploid and can be useful in further breeding programs. However, there are some general issues (i.e., reference citations and formatting are not as per the journal specification, and plants were not compared under stress as importance was highlighted on several occasions).  In addition, some other specific issues are noticed in writing that I have highlighted below. These need to be addressed appropriately. Also, some suggestions are provided that will be useful to improve the presentation of the manuscript.

 L 15-16: Italicize the botanical name

L 16: planted or cultivated?

L 19: ‘metabolic’ skipped, so also include it with morphological, physiological changes

L 19: replace 'this study' by 'the present study' as it confuses if this statement pertains with the previous study.

L 23: without mentioning about testing under stress, its mere speculation. Rather, you may say 'may possess better (not improved) stress tolerance ability and vigor

L 23: without mentioning about testing under stress, its mere speculation. Rather, you may say 'may possess better (not improved) stress tolerance ability and vigor

L 23: include 'non-targeted'  metabolites

L 26: C. reticulata (Italic)

L 31: Delete word 'introduction' from keyword

L 36: Citation of references are not as per the journal format; please do it in revision.

L 86, 88, 92, 94, 96: Itlicize the C. reticulata.

Also, change accordingly throughout the manuscript.

L 92-97: Authors have straightway stated what they did in this study, and later, how this study will be useful? But, it is better first mention the research gap that prompted to conducted this study with clearcut hypothesis and objective to perform the study.

L 108: Whether 4th or 5th leaves were counted from the apex of shoot or from bottom?

L 116: replace levels with contents

L 213:  you mean in table 2?

L 232: Figure is not readable, so increase their size including text/digits on x and y axis. Also, assign A, B,C.....to each of the physiological and biochemical indices with providing suitable caption.

L 369: it is better to write 'better' in place of 'improved'...than 2X

L 370: without comparing the performance of 4X with 2X under stressful versus control conditions, how you can state 4X showed improved stress tolerance?

L 385: Conclusion shall be in separate heading.

L 385: Insert tetraploid within the bracket after Autopolyploid..

L 411: References are not as per the journal style, so their setting must be done accordingly

L 15-16: Italicize the botanical name

L 16: planted or cultivated?

L 19: ‘metabolic’ skipped, so also include it with morphological, physiological changes

L 19: replace 'this study' by 'the present study' as it confuses if this statement pertains with the previous study.

L 23: without mentioning about testing under stress, its mere speculation. Rather, you may say 'may possess better (not improved) stress tolerance ability and vigor

L 23: without mentioning about testing under stress, its mere speculation. Rather, you may say 'may possess better (not improved) stress tolerance ability and vigor

L 23: include 'non-targeted'  metabolites

L 26: C. reticulata (Italic)

L 31: Delete word 'introduction' from keyword

L 36: Citation of references are not as per the journal format; please do it in revision.

L 86, 88, 92, 94, 96: Itlicize the C. reticulata.

Also, change accordingly throughout the manuscript.

L 92-97: Authors have straightway stated what they did in this study, and later, how this study will be useful. But, it is better first to mention the research gap that prompted to conduct of this study with a clear-cut hypothesis and objective to perform the study.

L 108: Whether the 4th or 5th leaves were counted from the apex of the shoot or from the bottom?

L 116: replace levels with contents

L 213: Do you mean 'in' table 2?

L 232: The figure is not readable, so increase their size including text/digits on the x and y-axis. Also, assign A, B,C.....to each of the physiological and biochemical indices by providing a suitable caption.

L 369: it is better to write 'better' in place of 'improved'...than 2X

L 370: without comparing the performance of 4X with 2X under stressful versus control conditions, how you can state that 4X showed improved stress tolerance?

L 385: Conclusion shall be in a separate heading.

L 385: Insert tetraploid within the bracket after Autopolyploid..

L 411: References are not as per the journal style, so their setting must be done accordingly

Best

Reviewer 2 Report

 Reviewer's report

The current study evaluated the morphological and physiological differences between the tetraploids and diploid ‘Xinyu mandarin’ plants. The objective was to assess if the morphological and physiological differences between the two ploidy levels could be advantageous in terms of adaptation towards abiotic stress. The tetraploids exhibited larger and thicker leaves with larger stomata and lower stomatal density. The activities of the key enzymes in photosynthesis, total chlorophyll content, phenolic and proline contents were also higher in tetraploids than in diploids. The overall difference in the proportion of primary and secondary metabolites followed the same pattern, indicating that the tetraploids may exhibit improved stress tolerance than diploids. This means tetraploids can serve as a good parent material for improvement of Citrus varieties against abiotic stress.

General comments

This study is of highly significant value, and it will be beneficial in both sectors, i.e. commercially, and citrus research industry.

Specific comments

Line 15 – Citrus is a genus name for of various Citrus fruits, but in the first line it appears as if Citrus is a fruit, please rephrase accordingly.

The line of problem statement is missing in the abstract, thus it is not clear why this study was conducted

Line 86 – “Citrus reticulata” is a scientific name, it must be Italicised. Apply this comment throughout the document.

Line 87 – “The ripe fruit is brightly coloured and has good texture and taste, because...” Re-attempt to explain the reason/s behind the mentioned desirable traits. The ability of the crop to adapt across the province is not a sound reason.

Line 91 – “In a previous study..” Please cite this study accordingly

Materials and Methods

Lines 100-102 – “Diploid Citrus reticulata (C2X)…” Add country and city at the end of this sentence.

Add the climatic data, i.e. annual rainfall, RH, min, max and average temperatures

Lin 108 – The leaves were collected in how many seedlings? Which experimental design was adopted in the layout?

Line 110 – How many leaves were used for ploidy analysis? This must be clear

Line 112 – How many leaves were sourced per branch and per tree? Remember: the number of trees must be a representable sample size, because there is a significant tree-to-tree variation.

Lines 124 – 131 – the methods are too summarized, in the way that doesn’t even reveal the processing method of the leaves. Were the leaves used as fresh or they were lyophilized?  How much of the leaf sample was used to quantify each of the indices? This section needs to be re-written and reveal all the details.

Results

Line 206 – Fig 1- Please increase the font of Fig axes and labels to atleast 12 points. Also add the p-value and LSD to Fig 1c and 1d, so that it will be easy to see if there was a significant difference between the performance of 2X and 4X.

Line 212 – “It was…” Start this sentence with “The functional components..”

Line 224 – Insert “attribute”. Stress tolerance is an important quality attribute..”

Line 227 – Replace “levels” with “contents

Lines 230 – 232 – Fig 2: All the labels in the graphs are not visible. Increase a font size to atleast 12 points for all the axes and labels in the graphs. Also add the p-values and LSDs for each graph in order to indicate if the difference was significant between 2X and 4X in all parameters.

Line 237 – “Compared to the diploid…” From the graph, I agree, it was higher, but was the difference significant? This was not indicated or explained where I see stars.

Line 270 - 290 – Fig 3 and 4 increase font size in graphs labels, they are not visible

Line 351 – Replace “smaller” with “lower

Line 361 – Make it “Discussion and Conclusion”

Add recommendations at the end, answering a “so what?” question

Reviewer 3 Report

In this manuscript by Wang et al., the authors did physiological and metabolic analyses of tetraploid and diploid Citrus reticulata (‘Xinyu mandarin’), and suggested the tetraploid may be more tolerant to stress, and better for growth. However, the manuscript has not been carefully prepared.

Major comment:

Line 3: the first and last names of authors seem to be flipped.

Line 15: "Citrus reticulata" should be italic

Line 194: "0.01 or 0.05", does it mean p-value?

Line 206-208 (Fig.1): Please the panels in detail. For panels A and B, how to distinguish it is 2x or 4x, and what is the peak around 25 in (B)? For panel C, how was the leaf index measured? Please describe that in the Method. For panel D, to label the statistics, the authors should draw a line above two bars and label  asterisks above the line. What do asterisks stand for? Please describe it in the legend or method.

Line 228-230: the logic here is not clear. What compounds is responsible for short-term or long-term environmental changes?

Line 232 (Fig.2): the legend keys are unnecessary. The information has already been provided in the title of y-axis. The asterisks need to be described, as suggested before.

Line 271 (Fig.3): In the panels B and C, what do "CK-1" and "S-1" stand for? What is the difference between panel B and panel A in fig.4?

Line 346 (Table.3): Table.3 is as the same as table.2. Please provide the right table.

Line 350-352: How do down-regulated flavonoids support "light factors affect polyploids to a smaller extent than other environmental factors"? 

Reviewer 4 Report

Why the experiment was conducted and how it will benefit the farmers should be clearly written in the objectives section.

Why the fourth and fifth leaves are collected for samplings?

It is written that each experiment was conducted thrice. How it will be authenticated?

For the analysis of chlorophyll generally, acetone is used in place of acetonitrile. Please check.

All the methodologies are not in detail.

The quality of Figures, 2, 3C, 4b, c, d should be improved.

A separate conclusion section is missing.

The quality of English is poor and should be checked by English-speaking colleagues.

Reviewer 5 Report

1. In the Introduction section: please add more information related to autopolyploid in Citrus (techniques which use to develop autopolyploid, advantages, disadvantages) from the previous studies.

2. In the Methodology section: it is better to add methodology for developing tetraploid plants (C4X).

3. In Figure 2. : please use bigger size of the font.

4. In the Discussion section: please add discussion related to molecular changes in tetraploid plants (information from references) and correlated to your results (morphological, physiological, and metabolic changes). 

Round 2

Reviewer 1 Report

The manuscript has been revised in view of comments made in the first round of revision. In this, authors have clubbed Results & Discussion which is logical as in the previous original version discussion was inadequate and very brief.

Hence, the authors need to include inferences drawn from their findings duly supported by others by omitting sub-headings as indicated/ highlighted in the clear PDF file attached herein for achieving the presentation to an acceptable level.

Reviewer 3 Report

The authors have improved a lot in this version. I am convinced by their reply.

Reviewer 4 Report

The authors have substantially revised the manuscript in light of the comments. Therefore, the manuscript can be accepted on the recommendation of editor.
